# Detection the eDNA of *Batrachuperus taibaiensis* from the Zhouzhi Heihe River Using a Nested PCR Method and DNA Barcoding

**DOI:** 10.3390/ani12091105

**Published:** 2022-04-25

**Authors:** Hongying Ma, Han Zhang, Jie Deng, Hu Zhao, Fei Kong, Wei Jiang, Hongxing Zhang, Xianggui Dong, Qijun Wang

**Affiliations:** 1Shaanxi Key Laboratory for Animal Conservation, Shaanxi Institute of Zoology, Xi’an 710032, China; mhying7916@163.com (H.M.); hanhanr9@163.com (H.Z.); dengjie0311@ms.xab.ac.cn (J.D.); zhaohu2007@126.com (H.Z.); k.coffee@163.com (F.K.); jiangwei197981@163.com (W.J.); zhs@ms.xab.ac.cn (H.Z.); 2College of Animal Science and Technology, Northwest A&F University, Xianyang 712100, China

**Keywords:** *Batarchuperus taibaiensis*, eDNA, genus-specific primer, nested PCR, DNA barcoding

## Abstract

**Simple Summary:**

The Taibai stream salamander (*Batrachuperus taibaiensis*) is a protected species endemic to the Zhouzhi Heihe River. Few traditional studies have been conducted on the distribution of *B. taibaiensis*, leading to irregular discoveries. Environmental DNA (eDNA) is an ideal species detection technique that can increase accuracy and decrease the cost of population surveys. Here, we have established an optimal method for obtaining the eDNA of *B. taibaiensis* from water samples, which provides a theoretical basis and reference for resource investigation and protection of *B. taibaiensis.* Moreover, this study highlights the detection of a rare species in a river for use in further research.

**Abstract:**

The Taibai stream salamander (*Batrachuperus taibaiensis*) is a recently described species of the genus *Batrachuperus* that occurs in the Zhouzhi Heihe River and is endangered in its native range. Here, we have established a method for water environmental DNA (eDNA) analysis of *Batrachuperus* using a series of optimizations. We have designed a specific set of primers for the genus *Batrachuperus* to amplify a 160 bp fragment of *Cytb*. The sequences were obtained from nested PCR on eDNA extracted from water samples, after which DNA barcoding was performed according to sequence analysis to determine the presence of the target species in the water. The method was validated using water from the Zhouzhi Heihe River with known *B. taibaiensis* populations and found that *B. taibaiensis* eDNA can move at least 150 m downstream from its point of origin. This study is the first to establish an optimal method for obtaining the eDNA of *Batrachuperus* from water samples, which provides a theoretical basis for resource investigation and the protection of *B. taibaiensis* in future research. It is also an example of the eDNA extraction of other species that live in similar waters and are less genetically diverse between species.

## 1. Introduction

Stream salamanders of the genus *Batrachuperus* (family *Hynobiidae, Urodela, Amphibia*) mainly live in clear water streams and prey on small shrimp and aquatic insects. The breeding season of *Batrachuperus* is March to April, and the eggs are laid in crevices where they live [1]. Stream salamanders, used in traditional Tibetan medicine in China, are used to treat traumatic injuries and joint pain. Due to their unique medicinal value, humans have begun to overexploit them, leading to a sharp decline in their population. Therefore, these species have been listed as endangered species [1]. To date, seven species of the genus *Batrachuperus* have been described in China: *B. cochranae*, *B. karlschmidti*, *B. londongensis*, *B. pinchonii*, *B. taibaiensis*, *B. tibetanus*, and *B. yenyuanensis* [1,2]. 

There are three types of *Batrachuperus* distributed in Shaanxi Qinling: *B. pinchonii*, *B. tibetanus*, and *B. taibaiensis* [1]. *B. taibaiensis* is a new species of the genus *Batrachuperus* that was found in the Zhouzhi Heihe River by Song Mingtao in 2001 [3]. The Heihe River rises from Er Ye Sea and Yu Huang Chi of Taibai Mountain in Zhouzhi Country and is characterized by rich plant and animal biodiversity [4]. The altitude of the Heihe River is 1200–3650 m, which is suitable for the survival of *B. taibaiensis* [5,6].

To date, research on the distribution of *B. taibaiensis* was conducted by Liang (2010) by combining morphology and molecular biotechnology to perform a statistical analysis based on the collected samples of *B. taibaiensis* [7]. However, these traditional methods depend on specific expertise and are difficult to standardize, which can create unknown rates of false absences in populations. Environmental DNA (eDNA) is an ideal species detection method that can increase the accuracy and decrease the cost of surveys. eDNA is any DNA that is freely present in the environment in the form of mucus, feces, skin cells, or gametes from an organism [8,9]. The study of aquatic species using eDNA has focused on vertebrate species, such as fish and amphibians [10,11,12,13], and has many applications, such as surveys in conservation biology [11,14,15], investigation of species invasion [12,13,16,17,18,19,20], researching species diversity and distribution [21,22], and estimating biomass measurements in a specific environment [23]. In the past decade, eDNA as a useful tool has seen remarkable interest in targeted species detection and biodiversity assessments [18].

The analysis of water eDNA involves a series of steps, including water eDNA capture, preservation, extraction, amplification, and sequencing to ensure a match with the target species. There are two types of water eDNA capture methods based on water quality and test conditions: direct freezing of water samples and freezing of filter membranes after filtration [17,21,24]. As for the preservation method, Ma (2021) compared common storage methods and found that a membrane directly soaked in alcohol is the best storage method that can increase eDNA yield. For eDNA extraction, Qiagen’s DNeasy Blood and Tissue Kit or the PowerWater DNA Isolation Kit were used as common commercial DNA kits [13,14,15,16,20]. In some cases, the phenol chloroform-isoamyl alcohol (PCI) extraction method was used to extract eDNA [22]. Conventional PCR and qPCR were conducted using a specific primer to identify the target species; the former sequences the target band and compares it with the sequence in the NCBI nucleotide repository to determine whether it is the sequence of the target species [10,11,19,25]; the latter uses the amplification curve to estimate the existence of the species or estimate how many species exist through copy number [12,26]. The nested PCR method is also a good choice when the concentration of aquatic species is low [27]. In the process of species identification, when the taxonomic differences between species are small, the classification of these species is conducted by DNA barcoding and phylogenetic analysis [20,28,29,30].

Here, we combined and optimized a series of eDNA steps to detect *B. taibaiensis* in the Heihe River and plan to use this method for long-term monitoring of the habitat of *B. taibaiensis* in the Heihe River. 

## 2. Material and Methods

### 2.1. Field Sampling and Collection

The Heihe River (Zhouzhi County, Shaanxi Province) originates from the Taibai Mountain National Nature Reserve. It is 125 km long, has a drainage area of 2258 km^2^ [31], and has good water quality and rich fishery resources (e.g., *Brachymystax lenok tsinlingensis, Andrias davidianus, Batrachuperus,* etc.) [4,32]. There are many tributaries along the Heihe River, such as the Ban Fangzi, Hua Erping, San Cha, and Miao Gou Rivers. Water samples were collected from two locations (San Cha River and Miao Gou River) (Figure 1). Sites were sampled at two time points: 4 April 2021 and 1 August 2021. In April, the water samples were collected from six sites within the San Cha and Miao Gou Rivers, and in August, from five sites at each of the two locations. The coordinates of each site and the distance between sites are shown in Table 1 and Figure 2, respectively.

We filtered 2 L of surface water at each site through 0.45 µm cellulose nitrate filters (disposable filter funnel, 47 mm gridded filter, Thermo product no. 145-2405), and precautions were taken not to contaminate the water sample by wearing gloves during filtering. Two filters were used for each site, and 2 L of distilled water was filtered as a negative control for each sample. Each filter was preserved in 95% ethanol in a separate 2 mL cryopreservation tube, stored on ice in the field, returned to the laboratory, and stored at −80 °C until DNA extraction.

### 2.2. Batrachuperus Genus Specific Primers for Nested PCR

Candidate primer sets were designed using default parameters in the Primer 5 software. To obtain sufficient PCR products and a clear band consistent with the expected amplicon size from the field water sample, a nested PCR strategy was used. We designed a new *Batrachuperus* genus-specific primer pair to nest the Ma (2021) primers inside its amplification product, which amplified a fragment of 160 bp in the *Cytb* gene. Amplification tests were performed to validate the new *Batrachuperus* genus-specific primers in vitro. The primer pairs used in this study are listed in Table 2.

### 2.3. Environmental DNA Extraction and Detection

We extracted DNA from the tissue of *Batrachuperus* species and optimized the primer pair PCR annealing temperatures using the protocols described in Ma (2021) [33].

We used a DNeasy Blood and Tissue Kit (Qiagen, Dusseldorf, Germany) to extract the eDNA from the filter. First, we removed the filters from the ethanol and air-dried them for about 4–5 h. Next, we divided each filter in half with a sterile razor blade and transferred each of them to 2 mL extraction tubes, then added 500 mL ATL buffer into every tube and cut the filter membrane into tiny pieces, and then digested the content of every tube with 30 μL Pk for approximately 48 h at 56 °C.

Referring to the instructions of the kit, we made the following adjustments: 400 mL of AL and 400 mL of absolute ethyl alcohol were added to each tube. Negative filters were used as controls for eDNA extraction to confirm that contamination did not occur during the process. The DNA concentration was assessed using a Nanodrop 2000 spectrophotometer (Thermo Fisher Scientific, Waltham, MA, USA).

PCR amplification of *Cytb* using the P2 primer described by Ma (2021) was performed using the following protocol. The amplification reaction was performed in a total volume of 25 μL, including 12.5 μL high fidelity Mix (Solarbio, Beijing, China), 9 μL double distilled water (ddH_2_O), 0.5 μL of each primer with final concentration, and 2.5 μL of template DNA. PCR conditions were as follows: an initial denaturation step at 94 °C for 3 min, 50 cycles of 94 °C for 10 s, annealing at 56 °C for 10 s, and elongation at 72 °C for 10 s. The final elongation step was performed at 72 °C for 3 min. The polymerase chain reaction was used as the template for nested PCR. Amplification of a smaller fragment of the *Cytb* gene with the P1 primer was performed using the same PCR conditions as those mentioned above. Ten µL of the PCR products were visualized on 2% agarose gel at 120 V for 20 min with 2 µL of Nucleic Acid dye (Solarbio, Beijing, China). Amplicon size was determined using a DM2000 ladder (Solarbio, Beijing, China).

### 2.4. Measures for Avoiding Contamination in eDNA

Throughout this study, we used separate rooms for DNA extraction, eDNA extraction, PCR, and post-PCR procedures. During eDNA extraction, we used different clean foam boxes as simple laminar flow hoods when drying each filter. Meanwhile, we ran all PCR and nested PCR reactions in four duplicates to discard false-positive and false-negative results due to technical failures. When the amplification results were not clear (the bands were too weak), the amplification was repeated using 3.5 µL of the PCR template instead of 2.5 µL as described above.

A minimum of two positive results from each site were considered valid to corroborate the presence of a species’ DNA in the sample, and we chose two positive PCR products for Sanger sequencing (Tsingke Biotechnology Co., Ltd., Beijing, China).

### 2.5. DNA Barcoding

The reference sequences for *Batrachuperus* were downloaded from GenBank. The sequences obtained in this study and the additional reference sequences were aligned using ClustalW in MEGA 7 [34]. The aligned sequences in our study were compared for their similarity to those in the NCBI GenBank database (http://www.ncbi.nlm.nih.gov, 15 January 2022) using the basic local analysis search tool (BLAST). The highest similarity of the queried sequence with the database sequences was determined, and the sequences that had a 98–100% similarity with the database sequences were identified as the respective species. MEGA 7 software was used to calculate the interspecific genetic distance and intraspecific genetic distance based on the Kimura-2-parameter (K-2-P) two-parameter model [29,34].

An ML tree was built using MEGA 7 software [34]. The HKY + I + G substitution model was selected using the jModel Test 2.1.4 [35,36]. The other parameters were the default values.

## 3. Results

### 3.1. Total eDNA Yield in Different Seasons

In this study, we successfully extracted eDNA from water samples from the Zhouzhi Heihe River using the DNeasy Blood and Tissue Kit (Qiagen, Dusseldorf, Germany) with optimized steps. The results showed that eDNA concentrations were higher in the August collections, especially in samples from the Miao Gou River (Appendix A).

### 3.2. Species-Specific Nested PCR

PCR was performed using the P2 primer [33] and high-fidelity mix, and four replicates were performed for each site (except for site 4-1-1 with six replicates). This PCR product was used as the template for nested PCR, and the nested PCR provided a clear band of 160 nucleotides (except for the negative controls). The nested PCR with at least two single clear bands from each site was considered valid in both the San Cha and Miao Gou River samples (Figure 3). Single clear bands were sequenced by nested PCR at Tsingke Biotechnology Co., Ltd. (Beijing, China).

According to the PCR and nested PCR results, we found that the band intensities from the samples collected in April tended to be brighter than those from August, suggesting a higher concentration of *Batrachuperus* eDNA in April (Figure 3). We also found clear bands in which the water sites were at least 50 m and 150 m downstream from the source population in the river (Table 1 and Figure 3).

### 3.3. Detection of Batrachuperus taibaiensis DNA in Water Sample

In this study, we chose two clear single bands for every collection site on the Heihe River in April and August (44 sequences in total). Sequence alignment was conducted between the sequences in this study, and *Batrachuperus Cytb* sequences were downloaded from NCBI, followed by truncation of the aligned sequences. The sequences of the amplified nested PCR products were deposited in the NCBI GenBank database with accession numbers OL351444-OL351487.

The *Cytb* gene sequences obtained in this study were compared with those available in the GenBank (*Cytb*) database. Each species showed high values of intraspecific homology, and 98–100% of them were among the *Batrachuperus* species (Appendix A).

In the *Cytb* sequences, the interspecies genetic distances among all *Batrachuperus* species were between 0.0532 and 0.1077 (Table 3), which were greater than the minimum species identification value of 0.020 suggested by Hebert [29]. The intraspecies genetic distances were between 0.0019 and 0.0322 (Table 4). The interspecies value was larger than the intraspecies value, indicating that *Cytb* can be used as an effective barcode gene for the accurate identification of *Batrachuperus* species. Meanwhile, the interspecies genetic distance between our sequence and *B. taibaiensis* from NCBI was the smallest. Therefore, we can speculate that the sequences in our study belong to *B. taibaiensis*.

A molecular phylogenetic tree was constructed using the Maximum Likelihood Method (Figure 4). *B. pinchonii, B. tibetanus,* and *B. taibaiensis* sequences from the NCBI database were clustered separately in the evolutionary tree. Sequences in our study were clustered with *B. taibaiensis*, indicating that *Batrachuperus* inhabiting the Heihe River belongs to *B. taibaiensis*. The phylogenetic tree was consistent with the results of the DNA barcoding analysis.

## 4. Discussion

This study is the first to demonstrate that eDNA methods can provide a basis for investigating the presence of *B. taibaiensis* in natural river habitats. *B. taibaiensis* is a vitally important and rare endemic species in the Qingling Mountains, and the application of eDNA techniques requires a series of optimized steps [37]. As shown in previous studies, a high-fidelity enzyme mix can better amplify the target band of water sample eDNA [33], and nested PCR is reliable when organisms are secretive or live at low densities [27]. Therefore, a high-fidelity enzyme mix and nested PCR method were used in our study to amplify the eDNA of *Batrachuperus*. Since the eDNA fragment of *Batrachuperus* was amplified using genus-specific primers, DNA barcoding and phylogenetic analysis were performed to determine the classification of the different species [20,28,29,30]. The results showed that the sequences obtained here had the smallest genetic distance from the *B. taibaiensis* sequence downloaded from NCBI and were also on the same branch as *B. taibaiensis* in the phylogenetic tree. It is necessary to increase the number of samples in the field and replicate more molecular workflows to enhance the reliability of the eDNA analysis of rare species [14,25]. This was done in our research. Studies on the distribution of *B. taibaiensis* have shown that this species is mainly located in the upper reaches of the Zhouzhi Heihe River basin [5,38]. Our research also revealed that the species of *Batrachuperus* inhabiting the Heihe River belonged to *B. taibaiensis*.

Our study showed that overall, the total eDNA concentration in August was higher than that in April in the Heihe River (shown in Appendix A), whereas band intensities of the eDNA concentrations of *B. taibaiensis* were higher in April than those in August (Figure 3). Therefore, we can infer that the total eDNA concentration is not proportional to the target eDNA concentration. This can be explained by factors that influence the eDNA concentration of the target species. More populous species had a higher rate of detection and concentration of eDNA [12]. In our study, the samples of *B. taibaiensis* collected in April were found at intervals of 20–30 m, whereas in August, the intervals were 50–100 m. We obtained a high concentration of *B. taibaiensis* eDNA in April. We also need to take into account the sampling season and physiological conditions of the animals [11,39]. Unlike rainy August, April with warm temperatures is also the breeding season for *B. taibaiensis* [5,40], which leads to increased mobility and metabolism of *B. taibaiensis* and more DNA released into the environment through skin excretions, sloughed cells, and mucus excretion.

We detected *B. taibaiensis* eDNA in the stream where they live; however, the extent to which the target eDNA can be transported downstream through a flowing stream is unknown. We collected eDNA samples 50 and 150 m downstream from the source population of the river. The first reason for these choices is that the electricity required for the filtering device is often unavailable in the field; therefore, the number of filtered samples is limited. Another reason could be related to the results of previous studies. As both salamanders and *B. taibaiensis* live in cool, shady stream conditions, their transportation distances may be similar. Pilliod (2014) collected eDNA samples from caged salamanders at 5 m and 50 m downstream of the original source [41]. Although the eDNA of salamanders was only detected at 5 m, the eDNA of *B. taibaiensis* was also detected at 10 m (Table 1). Therefore, we wanted to identify whether the eDNA of *B. taibaiensis* could be collected at 50 m, and the answer was yes. The reason for this phenomenon may be that the caged salamanders had some stress response when placed in the river, whereas *B. taibaiensis* remained calm in the water. In other words, if there is no frequent movement of individuals, the degradation rate of eDNA will decrease [41]. Eichmiller (2014) [12] and Jane (2015) [42] detected target fish eDNA at 100 m and 239.5 m downstream, respectively. Because fish have a higher detection rate, we chose 150 m as the longest distance from the source location of *B. taibaiensis*, and eDNA could also be detected at this distance. In the future, we can extend the distance to obtain more information about the transport distance of *B. taibaiensis* eDNA because eDNA may travel much further distances, possibly in the order of kilometers [8]. In summary, *B. taibaiensis* eDNA can reach at least 150 m downstream from its point of origin, and we suspect that the density of the population has a significant influence on eDNA transport distance.

## 5. Conclusions

We have demonstrated that *B. taibaiensis* eDNA from the Heihe River can be amplified using genus-specific primers by nested PCR and barcoding sequence analysis. *B. taibaiensis* eDNA can be obtained at least 150 m downstream from its point of origin. This approach can now be applied for the detection of other rare species in rivers.

## Figures and Tables

**Figure 1 animals-12-01105-f001:**
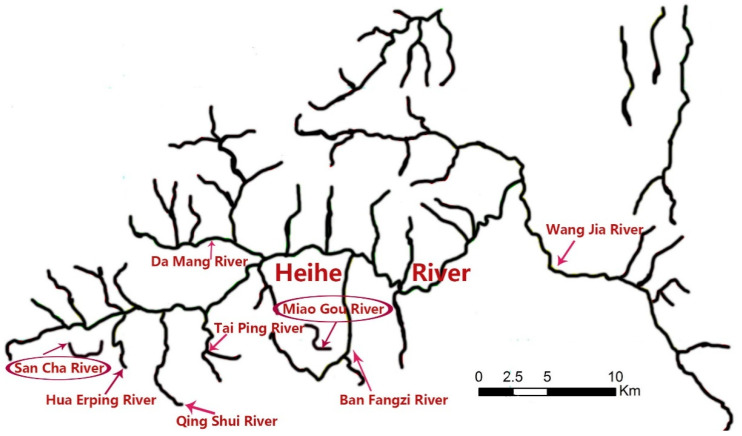
Geographic area of partial Heihe River (red circles represent locations where water samples were collected in our study).

**Figure 2 animals-12-01105-f002:**
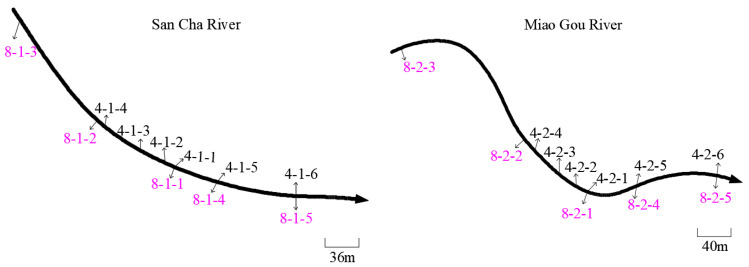
Schematic diagram of sampling sites within San Cha River and Miao Gou River (numbers in black represent the data of April collection, and the numbers in purple represent August collection; arrow indicates the direction of river).

**Figure 3 animals-12-01105-f003:**
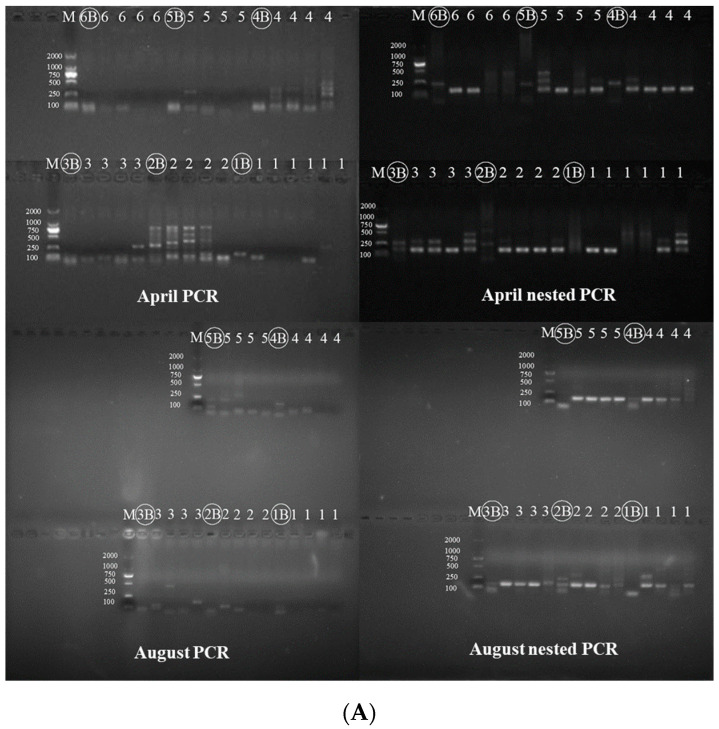
PCR and nested PCR results of water eDNA sample for San Cha River (**A**) and Miao Gou River (**B**). (M, marker; B, blank; 1-6: last number of site name in Table 1).

**Figure 4 animals-12-01105-f004:**
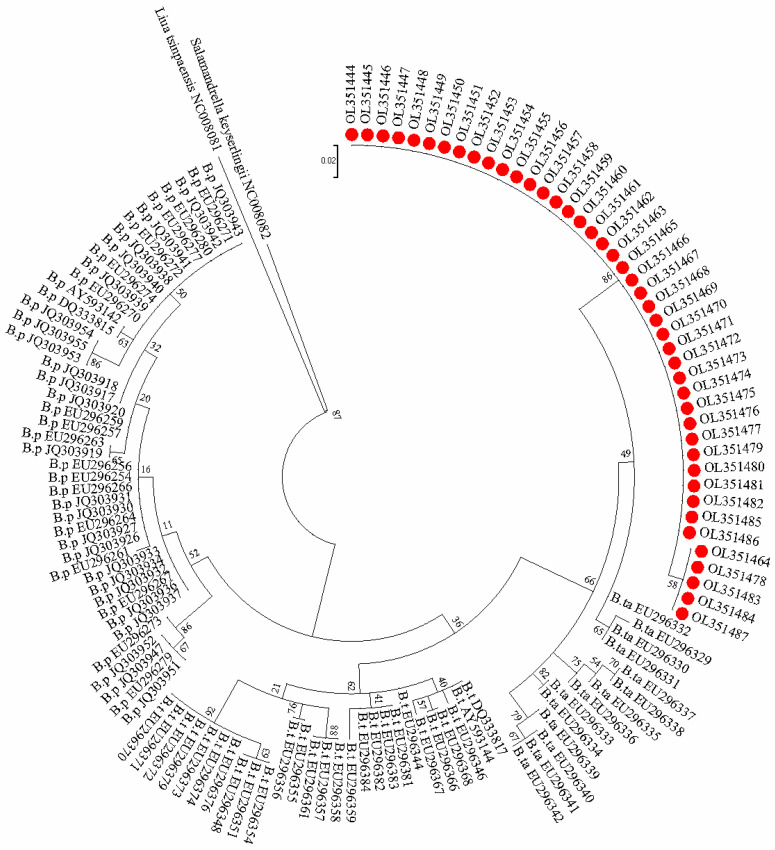
Molecular phylogenetic tree by Maximum Likelihood method (The ML tree is based on the HKY + I + G model; B. p, *Batrachuperus pinchonii*, B. t, *Batrachuperus tibetanus*, B. ta, *Batrachuperus taibaiensis*; red circles show the sample sequences from this study; *Liua tsinpaensis* and *Salamandrella keyserlingii* as an outgroup; numbers on branch in the tree are bootstrap values with percentage.).

**Table 1 animals-12-01105-t001:** The information on each site where we collected water for eDNA analysis (The site where the *Batrachuperus* species appeared when we collected water samples is shown with “+”, otherwise with “−”).

Collection Time	Location	Site Name	Coordinates	Distance between the Most Upstream Site and the Other Site (m)	*Batrachuperus* Species Appeared or Not
4 April	San Cha river	4-1-4	33.82884 N, 107.81094 W	0	+
		4-1-3	33.82897 N, 107.80912 W	50	+
		4-1-2	33.82917 N, 107.81048 W	80	+
		4-1-1	33.82938 N, 107.81062 W	90	+
		4-1-5	33.82890 N, 107.81094 W	140	−
		4-1-6	33.82918 N, 107.81126 W	240	−
	Miao Gou river	4-2-4	33.82928 N, 107.95356 W	0	+
		4-2-3	33.82919 N, 107.95385 W	40	+
		4-2-2	33.82918 N, 107.95362 W	65	+
		4-2-1	33.82920 N, 107.95325 W	80	+
		4-2-5	33.82859 N, 107.95459 W	140	−
		4-2-6	33.82858 N, 107.95496 W	240	−
1 August	San Cha river	8-1-3	33.82899 N, 107.80912 W	0	+
		8-1-2	33.82837 N, 107.80970 W	150	+
		8-1-1	33.82951 N, 107.81059 W	250	+
		8-1-4	33.82928 N, 107.81107 W	300	−
		8-1-5	33.82932 N, 107.81181 W	400	−
	Miao Gou river	8-2-3	33.82969 N, 107.95307 W	0	+
		8-2-2	33.82942 N, 107.95377 W	200	+
		8-2-1	33.82925 N, 107.95342 W	300	+
		8-2-4	33.82875 N, 107.95449 W	360	−
		8-2-5	33.82852 N, 107.95499 W	460	−

Note: 4-1-1, 4-2-1, 8-1-1, 8-2-1 are the source population sites in our study.

**Table 2 animals-12-01105-t002:** Primer pairs for partial *Cytb* gene amplification of samples from *Batrachuperus* genus.

Primer	Primer Sequence	Annealing Temperature (°C)	Size (bp)	Origin
P1	F:5′-GTAGATAAGGCTACTCTTACTC-3′ R:5′-ATGGGTGGAATGGAACT-3′	46	160	This study
P2	F:5′-TTGAGGTGGGTTCTCTGTAGATAAG-3′ R:5′-GGTTGGCGGGTGTAAAA-3′	52	290	[33]

**Table 3 animals-12-01105-t003:** The interspecies genetic distances of *Batrachuperus*.

	*B. pinchonii*	*B. tibetanus*	*B. taibaiensis*	Species in Our Study
*B. pinchonii*	/	0.0677 ± 0.0204	0.0963 ± 0.0266	0.1077 ± 0.0313
*B. tibetanus*	/	/	0.0928 ± 0.0257	0.0947 ± 0.0282
*B. taibaiensis*	/	/	/	0.0532 ± 0.0195

Note: Data are presented as interspecies genetic distance ± standard deviation.

**Table 4 animals-12-01105-t004:** The intraspecific genetic distances of each *Batrachuperus* species.

	*B. pinchonii*	*B. tibetanus*	*B. taibaiensis*	Species in Our Study
intraspecific genetic distances	0.0253 ± 0.009	0.0310 ± 0.0109	0.0322 ± 0.0113	0.0019 ± 0.0018

Note: Data are presented as intraspecies genetic distance ± standard deviation.

## Data Availability

The sequences data were deposited in the NCBI GenBank database with accession numbers OL351444-OL351487.

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
