# Peer review of "Detection the eDNA of Batrachuperus taibaiensis from the Zhouzhi Heihe River Using a Nested PCR Method and DNA Barcoding"

_animals, 2022, doi:10.3390/ani12091105_

Round 1

Reviewer 1 Report

Too many editorial revisions are needed.

  1. line 2: Batrachuperus taibaiensis should be italic
  2. line 14: Batrachuperus taibaiensis should be italic, and delete (B. tabaiensis)
  3. line 16: Environment  should be Environmental
  4. line 21: delete (B. tabaiensis)
  5. line 22; add acomma after Here
  6. line 26: add a space after ter
  7. lines 28, 31: add a space after B.
  8. line 41: delete "of" before decline     add a space before [1]
  9. lines 42, 43, 44, 45, 50, 52, 84, 85: add a space after B.
  10. lines 43, 45, 46, 48, 49, 60, 63, 67, 71, 73, 77, 78, 93: add a space [reference number(s)]
  11. lines 60, 67, 71, 75, 77, 81: [11, 14-15], [12-13, 16-20], [17, 21, 24], [13-16, 20], [10-11, 19, 25], [12, 26], [20, 28]
  12. lines 94-95: scientific names of animals should be italic
  13. lines 97, 101: Fig 1, Fig 2 should  be Fig. 1, Fig, 2
  14. line 95: [4, 30]    add a space before [4]
  15. lines 102104: add a space after 2, 0.45, 2
  16. line 139: add a space after new
  17. line 140: in vitro should be in vtro?
  18. Table 2: delete ( after temperature  and add (  before °C
  19. lines 148, 150: hours should be h
  20. lines 149, 151, 152: add a space before ml
  21. line 158: add a space after water
  22. line 164: add city, state, country after G8140
  23. line 165: Measures should be Measurements?
  24. line 173: add city, state, country after Ltd.
  25. lines 179, 180, 181: add a space before [reference(s)]
  26. line 185: environment DNA should be eDNA
  27. line 186: delete , Hilden, Germany
  28. line 188: high should be higher
  29. lines 195, 199, 210, 211: Fig should be Fig.
  30. line 198: more high should be higher
  31. lines 203, 211, 214, 215, 218, 219, 223, 226, 232, 237, 235, 238, 242, 246, 248, 250: Batracghuperus should be B.
  32. line 222: add the after is
  33. lines 228, 229, 231, 234, 237, 247, 249: add a space before [references
  34. line 231, 234, 237, 249: [20, 28], [14, 25], [5, 38], [11, 39]
  35. line 241: table should be Table
  36. line 246: add a space after eDNA
  37. lines 251, 274, 277: [5, 40], [12, 41]     add a space before [references
  38. lines 251, 254, 261, 263, 267, 270, 271, 277, 279, 281: Batrachuperus should be B.
  39. lines 256, 259, 260, 261 , 263, 264, 265, 266, 271, 274, 282: add a space after values
  40. lines 259, 265, 270, 277: add a space before (2014), (2014), before [8], [42]
  41. lines 259, 265: Pilliod (2014) should be [42], Eichmiller (2014) should be [12], Jane (2015) should be [41]
  42. References: too many revisions should be needed, just check the journa's instruction how to prepare references: titles should be not be large capitals, except the first word, journal names should be appropriately abbreviated , scientific names of animals should be italic add a space in appropriate places

Reviewer 2 Report

The proposed manuscript “Detect the eDNA of Batrachuperus taibaiensis from Zhouzhi Heihe river using nested-PCR and Phylogenetically Informative methods” introduces analysis of eDNA as a valuable method for species detection, especially for species with limited occurrence or threatened species. The whole laboratory analysis is based on two rounds of PCR which is a basic biological method, and the whole analysis can be completed in a very short time. Species identification can be easily done using BLAST algorithm and phylogenetic analysis has only an additional value. Thus the work is methodologically on a low level.

There are a lot of errors and some of them I consider as crucial ones which are causing the poor quality of the proposed manuscript. I will not mention all discrepancies specifically due to their high incidence. In general terms: Extensive English correction is needed, incorrect use of commas/dots in sentences, too many typos, grammatical errors, all these discrepancies make the manuscript reading difficult. There are redundant information in the introduction (e.g., why author listed methods/Kits for DNA extraction). Nested PCR is a suitably used method for the proposed manuscript but it is not explicitly explained why authors used it. What region was amplified using original primers designed by Ma et al (2021)? Are new primers for nested PCR species- (e.g., line 189) or genus-specific (e.g., line 134)??

The proposed ms is on the edge of rejection and major revision. After consideration of the manuscript quality I suggest a rejection and send it to another more specific journal.

Reviewer 3 Report

Animals

Manuscript ID: animals-1611466

Detect the eDNA of Batrachuperus taibaiensis from Zhouzhi Heihe river using nested-PCR and Phylogenetically Informative methods

The authors conducted eDNA surveys of the recently described Taibai stream salamander (Batrachuperus taibaiensis) in the Heihe River basin of China. They used a nested PCR approach with genus-specific primers and conventional PCR detection, with sequencing of PCR products to confirm the species identity. The data are useful, and the paper is well-written overall. I recommend publication with minor revisions. The following comments should be addressed by the authors:

Throughout, correct errors in English usage. The paper would benefit from a professional English editing service.

Title: I suggest shortening the title: “Nested PCR detection of environmental DNA from the rare Taibai stream salamander (Batrachuperus taibaiensis) in the Heihe River basin”

Simple summary, lines 14-15: I suggest editing the first sentence to: “The Taibai stream salamander (Batrachuperus taibaiensis) is an endemic protected species in Zhouzhi Heihe river.”

Abstract, lines 21-22: I suggest editing the first sentence to: The Taibai stream salamander (Batrachuperus taibaiensis) is a recently described species of genus Batrachuperus which occurs in Zhouzhi Heihe river, and is endangered in its native range.

Methods and Results: The Abstract and Discussion refer to the distance from the source population that allowed detection of B. taibaiensis eDNA, but the determination of the locations of source populations is not described in the Methods or Results. Please add this information, and add the locations of source populations to Table 1.

Methods: I found the site numbers confusing, I expected increasing site numbers from upstream to downstream, but this is not the case. To clarify, please make the font much bigger in Figure 2, and in Table 1, give the distance along the river from the most upstream site rather than the distance to the next site.

Table 1: Does the column labeled “Batrachuperus” show sites with positive eDNA detections? Please explain.

Methods, lines 168-169: I think “amplification” should be substituted for “digestion”, twice in this sentence.

Methods, line 171 and Results, line 194: The Methods says the threshold for a positive site was 2 replicates with clear bands, but the Results says it was 3 replicates with bands. Please clarify.

Results, lines 184 and 187: Change “eDNA yield” to “total eDNA yield”.

Figure 3: The labels are too small to read, and the bands are difficult to see. Please make everything bigger.

Results, lines 197-199: How was eDNA concentration determined? Band intensity does not always correlate well with starting concentration of the target sequence. This statement could be moderated: “We found that band intensities from samples collected in April tended to be greater than those from August, suggesting a higher concentration of Batrachuperus eDNA in April.”

Discussion, lines 241-242: Moderate to “while band intensities suggested that eDNA concentrations of Batrachuperus may have been higher in April than in August”

Discussion, lines 246-248: “In our study, samples collected in April, Batrachuperus taibaiensis were found at intervals of 20-30 m, while the intervals is 50-100m in August, in that way, we get high concentration of Batrachuperus taibaiensis eDNA in April.” I don’t understand how the site locations are connected to eDNA concentrations, please explain.

Discussion, lines 256-257: Again, please explain how the locations of the source populations were determined.

Discussion, lines 274-277: “While faster rates of flow would flush eDNA more quickly, this can explain salamanders eDNA samples not detect at 50 m, but when no diel change and no frequent movement of individuals, the degradation rate will decrease, current velocity should be relatively unimportant[42], Batrachuperus taibaiensis in our study confirm to this characteristic.” This sentence is not clear. Did you measure eDNA degradation rates?

Animal Ethics: “The animals involved in this study died naturally, other water samples do not involve ethics.” Please explain. What animals were involved in the study? The Methods only describes water samples. Did you collect carcasses from the rivers?

Reviewer 4 Report

The article describes the eDNA detection for an endangered species of the stream salamander in two tributaries of Heihe river in the North of China. Authors describe the procedure of collecting eDNA in detail (still it could be more detailed as it is important). Authors collected from the same site twice with a span of four months (the choice of months for collection could be justified better). Authors took many precautions as repeating amplification and counting only sites with two positive amplifications as the presence. This is nice side of the work.

Further authors have sequenced PCR product and aligned it to known B. taibaiensis sequences to confirm that it was indeed eDNA from the target species of the stream salamander. Further authors performed phylogenetic analysis (that was really unnecessary in this case and should be removed from the title as it only serves as a nice illustration for the species identification, that was actually done by BLAST, not by phylogeny).

I would suggest to work a bit on Figures (see comments below). The choice of figures is great, but the main figure with PCR data is terribly presented! So it is impossible to read the figure, analyse it and to review it. The article is well structured and the Introduction is nice, however, some very simple mistakes in English make it a bit hard to read. Please correct English a little bit (see some suggestions in the uploaded pdf). 

Introduction

Intor and abstract and title should contain common name for this salamander, not only Latin name!!

While description of the salamander and its use by humans is nice, it would be also nice to add a sentence or two about biological features of this species: what kind of waters it prefers, how it behaves through the year (does it migrate or stay in one place), what it feeds on, does it reproduce in the same spot or it goes elsewhere, when is reproductive season, what are other species of salamanders that might live in the same spot? A little bit of information needed here to be able to interprete the eDNA data that authors obtained and to understand the differences between April and August datapoints.

Materials and methods

About water sample collection –Did you stay downstream while collecting to avoid the contamination?

Annealing temperature – I am not sure how to deal with it,but why is annealing temperature for the specific nested primers are so low? 46C? In particular for nested specific primers it can be much higher in the upper 50th or low 60th to produce specific amplification. How can you consider amplification at 46C specific? I wish authors would provide explanation or references to similar works. The gel has some nice clear bands, so it must have been good amplification. But why choose a primer pair with such a low annealing temperature?

For eDNA it is critical to perform experiments in a certain cautious way to prevent contamination and crosscontamination. While authors write that three steps were performed in different rooms (great!), still some questions remain. Was the laminar hood used to prevent the contamination of eDNA? Especially while drying filters for 4-5 hours – the contamination can easily occur. What other measures were used to prevent contamination by lab amplicons or samples? Especially with the control DNA from a real salamander - was this DNA around while working with eDNA?

Gel description should contain more details: amount of DNA on the gel, voltage, duration of the run; DNA ladder.

The method of sequencing should be described (Sanger sequencing?)

BLAST of the sequences is a critical step in the species identification and should be described in more details – it should be a separate section in the M&Methods. This section should be called DNA barcoding. What reference sequences were used? What was sequence identity measure? What was the difference with the next closely related species of Batrachuperus? It is all important issues that often obscure results of species identification in eDNA analysis.

On the other hand, the phylogenetic tree – it has secondary role here in the paper, just for the illustration and does not need to be described in such a detail.

English

Avoid extra long sentences.

Once the full name of the genus and species was mentioned for the first time in the manuscript, you need to abbreviate the genus name to one letter and write it all the way through the manuscript.

Batrachuperus taibaiensis – for the first time in the paper B. taibaiensis – everywhere else in the paper.

genus and species names are always in italic. There is always a space between genus abbreviation and species.

The PCR is called “nested PCR”, please make it uniform through the text so there would be no nest-PCR, NEST PCR, etc.

Additional comments on English please see in the attached PDF. I how the editing feature will show my edits.. I used Adobe Acrobat Reader to make edits.

Results

it is really hard to judge results as the gel picture is unreadable.

Discussion

Overall the discussion is nice. Authors attempted to provide reasoning to some interesting phenomena they have revealed during the study.

There must be something wrong with Figure 2 or Table 1. It says that positive eDNA for T. taibaiensis was found 150 m DOWNstream, but the point is Upstream. Also there is no explanation for positive 200 m value ( point 8-2-3 in the Table 1).

I am not sure that the phylogenetic analysis is correct way to describe the procedure of the species identification by CytB sequence. Usually it is called DNA species barcoding and basically represents the value of similarity between reference sequence of the species in some taxanomic database and the investigated sequence. I agree that phylogenetic tree provides good visual representation of the fact that eDNA in this study indeed belongs to the B. tabaiensis. But this is more like a secondary representation while DNA barcoding data and BLAST values should be presented first.

Figures

Figure 1. This figure is very useful to understand the location of the collection. I would suggest joining this figure with Figure 2. This drawing can be made a bit smaller to fit on the left side of the figure 2. On standard maps the river names are written along the river line. If this is hard to do, then make tributarie’s name a bit larger.

Figure 2. This is very nice and useful figure. However, please make the arrows of the direction of rivers to look nicer. I would also suggest increasing font size of the location numbers.

Also, are you sure the direction of the flow is indicated correctly? It seems like the 150 m positive point is located upstream and not downstream like it says in the text.

Figure 3.

The gels have some nice bands! A lot of work is seen here through the necessary repetition of PCR with eDNA. Too bad that the quality of the gel picture is so poor!!

The red numbers and letters on this figure are not readable. Maybe use white color for better visibility. Also sample numbers can be coded simply by 1,2,3… and deciphered in the table. Or image should be in higher resolution to be able to read the designations.

The name of the marker (size ladder) must be provided.

It is not clear what top and bottom portion of the gel represents on each gel.

Maybe inverting the image so the bands become black color and the background becomes white color would be good way to make figure better for the readers.

I also could not understand – is the gel presented upside down? Usually the gel wells (pockets) are on top. If the gel is upside down please flip it back so it is represented in conventional way with gel wells on to and the larger size fragments higher than the lower size.

submitted sequences – I could not find sequences with accession numbers OL351444-OL351487 in the NCBI database? Are they not submitted yet?

For the phylogeny it would be better to use at least one outgroup sequences from some distant Batrachupterus species or species outside of Batrachupterus. It is a good practice to use an outgroup while building a tree of closely related species. Also it would be nice if bootstrap values would be displayed at the nodes of the tree.

Just by looking at the tree, I can not immediately understand why relatively conserved and such short fragment of the Cytb gene has so many differences even between individuals of B. taibaiensis? They all should have the same sequence, unless this region of Cytb is some kind of highly variable fragment or repeat.

Supplemental table.

Change table title to 

Table S1. Total eDNA concentration.

Change the column name from "Collect time of April/August" to "April collection" and "August collection."

I could not understand what “Black” means in the table? Is this a negative control? Blank control? No eDNA control? Please clarify.

Round 2

Reviewer 2 Report

The proposed manuscript “Detect the eDNA of Batrachuperus taibaiensis from Zhouzhi Heihe river using nested-PCR and Phylogenetically Informative methods” was improved and now it has a higher quality of presentation and English grammar. I disagree with the statement of DNA extraction kits in the introduction and I think it should be a part of the methods. It is not a methodological manuscript. Eventhow if authors are convinced that information is very important in the intro, it is only a detail which does not influence the quality of the paper.

After linguistic corrections, the manuscript can be accepted for a publication in the Animals journal. I recommend a minor revision and using the English Editing service.

Author Response

Dear Reviewer:

      Thank you very much for your suggestion about our manuscript. We are revising our manuscript based on your comments. The paper have been done the professional English editing service, while the statement of DNA extraction kits in the introduction is indispensable, so we want to keep it, hope you can understand.

Kind regrads,

Hongying Ma
